# Is Gross Extrathyroidal Extension to Strap Muscles (T3b) Only a Risk Factor for Recurrence in Papillary Thyroid Carcinoma? A Propensity Score Matching Study

**DOI:** 10.3390/cancers14102370

**Published:** 2022-05-11

**Authors:** Yongseon Kim, Yong-Seok Kim, Ja Seong Bae, Jeong Soo Kim, Kwangsoon Kim

**Affiliations:** Department of Surgery, College of Medicine, The Catholic University of Korea, Seoul 06591, Korea; 21600356@cmcnu.or.kr (Y.K.); dydtjr79@catholic.ac.kr (Y.-S.K.); drbae@catholic.ac.kr (J.S.B.); btskim@catholic.ac.kr (J.S.K.)

**Keywords:** papillary thyroid carcinoma, extrathyroidal extension, disease-free survival, propensity score matching

## Abstract

**Simple Summary:**

In papillary thyroid carcinoma (PTC), staging classification of gross and minimal extrathyroidal extension (ETE) has been recently modified in the eighth edition of the American Joint Commission on Cancer/Union for International Cancer Control (AJCC/UICC) TNM staging system. In this study, we compared the clinicopathological characteristics and recurrence rates between minimal and gross ETE. No significant differences in the recurrence and disease-free survival rates were found between the two groups. Whether gross ETE invading strap muscles (T3b) only could be a risk factor for recurrence in PTC remains questionable.

**Abstract:**

The presence of extrathyroidal extension (ETE) is associated with locoregional recurrence and distant metastases in papillary thyroid carcinoma (PTC). This study was designed to compare the recurrence risk between minimal ETE (mETE) and gross ETE (gETE) in patients with PTC using propensity score matching. In this study, 4452 patients with PTC who underwent thyroid surgery in a single center were retrospectively analyzed, and clinicopathological characteristics were compared according to the ETE status. Disease-free survival (DFS) and recurrence risk were compared between mETE and gETE after propensity score matching. The mean follow-up duration was 122.7 ± 22.5 months. In multivariate analysis, both mETE and gETE were not associated with recurrence risk before propensity score matching (*p* = 0.154 and *p* = 0.072, respectively). After propensity score matching, no significant difference in recurrence rates was observed between the two groups (*p* = 0.668). DFS of the gETE group did not significantly differ from that of the mETE group (log-rank *p* = 0.531). This study revealed that both mETE and gETE are not independent risk factors for the risk of recurrence in PTC. Our findings suggest that gETE invading strap muscles only might not be associated with worse oncological outcomes in PTC.

## 1. Introduction

Extrathyroidal extension (ETE) is a risk factor for prognosis in patients with papillary thyroid carcinoma (PTC) [1,2]. As defined by the American Joint Commission on Cancer/Union for International Cancer Control (AJCC/UICC), ETE can be classified into gross ETE (gETE), which is visually confirmed intraoperatively by surgeons, and minimal ETE (mETE), defined as tumor cells extending to strap muscles or perithyroidal tissue and confirmed by pathological review [3]. The diagnosis of mETE can be challenging because histopathological findings of mETE usually vary among pathologists. Recently, mETE was excluded from the T3 classification in the eighth edition of the AJCC/UICC TNM staging system [4]. gETE to strap muscles alone has now been classified as a T3b tumor, gETE to subcutaneous soft tissue, the larynx, the trachea, the esophagus, or laryngeal nerve is considered a T4a tumor, and invasion of the prevertebral fascia, the carotid artery, or mediastinal vessels is classified as a T4b tumor [4]. Therefore, this modification in the eighth edition of the AJCC/UICC TNM staging system has led to the downstaging of many patients [5]. However, the American Thyroid Association (ATA) management guidelines still consider the presence of mETE as a feature of an intermediate risk of recurrence, regardless of TNM staging modification [6].

The role of mETE as a risk factor for recurrence remains controversial. Some studies have compared the outcomes between no ETE and mETE, between mETE and gETE, or among the three groups together [7,8,9]. Danilovic et al. have reported that both mETE and gETE are independent risk factors for recurrence in PTC [10]. Park et al. have examined 381 patients with PTC and found mETE to be correlated with aggressive histopathological features and tumor recurrence, concluding that patients with mETE have poorer clinical outcomes than those without ETE [11]. In contrast, several studies evaluating differentiated thyroid carcinoma (DTC) with mETE without lymph node (LN) metastases have found no statistically significant increase in the risk of recurrence [12,13,14].

The assessment of ETE is a key factor not only in establishing the staging system but also in determining the patient’s surgical extent, adjuvant treatment, and the intensity of surveillance during follow-up. As mentioned earlier, several studies have compared the prognosis of each ETE group; however, most studies had selection bias due to their retrospective designs, making it difficult to reach significant conclusions.

Therefore, this study was designed to compare clinicopathological characteristics and long-term oncological outcomes among different degrees of ETE using propensity score matching analysis to reduce selection bias in patients with PTC. Moreover, we performed a sub-analysis to identify the clinical significance of ETE in patients with papillary thyroid microcarcinoma (PTMC).

## 2. Materials and Methods

### 2.1. Patients

We retrospectively reviewed 4591 patients with PTC who underwent thyroid surgery from March 2008 to June 2014 at Seoul St. Mary’s Hospital (Seoul, Korea). In total, 84 and 55 patients were excluded from the analysis because of insufficient data and loss to follow-up, respectively. The medical charts and pathology reports of 4452 patients were reviewed and analyzed. Of the patients, 1137 (25.5%) underwent lobectomy and/or contralateral partial thyroidectomy (less than total thyroidectomy (TT)) and 3315 (74.5%) underwent TT.

The mean follow-up duration was 122.7 ± 22.5 months (range, 92–167 months). This study was conducted in accordance with the Declaration of Helsinki (as revised in 2013). This study was approved by the Institutional Review Board of Seoul St. Mary’s Hospital, Catholic University of Korea (IRB No. KC22RISI0041), which waived the requirement for informed consent due to the retrospective nature of this study.

### 2.2. ETE Definition

According to the T stage classification based on the eighth edition of the AJCC/UICC TNM staging system, the T3b stage grossly invades strap muscles only, and the surgeon writes it on the operation record after confirmation during surgery [4]. Since the patients included in this study were admitted from 2008 to 2014, we referred to the pathology reports in that period. This is because the eighth edition of the AJCC/UICC staging system was revised in 2016. In this study, the gETE group included T3b, that is, invading strap muscles only, and excluded T4a or T4b.

mETE is defined as extrathyroidal invasion restricted to perithyroidal soft tissues, including microscopic strap muscle invasion [3]. The mETE group was classified according to this definition in this study.

### 2.3. Follow-Up Assessment

Postoperative care and follow-up were performed according to the ATA management guidelines [6]. For the follow-up, all patients underwent physical examination, serum thyroid function tests, measurement of thyroglobulin and anti-thyroglobulin antibody concentrations, and neck ultrasonography every 3–6 months for the first year and annually after that. Radioactive iodine (RAI) ablation was performed 6–8 weeks after TT using doses based on the ATA management guidelines, and whole-body scans (WBS) were performed 5–7 days after RAI ablation. During routine follow-up evaluation, patients with suspected recurrence underwent additional diagnostic imaging tests, including computed tomography, positron emission tomography/computed tomography, and/or RAI WBS, to determine the location and extent of recurrence. Disease recurrence was confirmed using imaging modalities and/or pathological diagnosis using ultrasound-guided fine-needle aspiration/core needle biopsy or a surgical biopsy specimen.

### 2.4. Primary and Secondary Endpoints

The primary endpoint was a comparison of disease-free survival (DFS) between the mETE and gETE groups after propensity score matching, and the secondary endpoint was a comparison of clinicopathological characteristics between the two groups before and after propensity score matching analysis.

### 2.5. Statistical Analysis

Continuous variables are presented as means with standard deviations, and categorical variables are reported as numbers with percentages. Student’s *t*-test was used to compare continuous variables. We compared the differences in categorical clinicopathological characteristics among the ETE groups using Pearson’s chi-square test or Fisher’s exact test. Univariate Cox regression analyses were performed to validate DFS predictors, and statistically significant variables were analyzed using a multivariate Cox proportional hazard model. Hazard ratios (HRs) with 95% confidence intervals (CIs) were calculated. DFS was compared using Kaplan–Meier survival analysis, and the log-rank test was used to calculate significant differences.

We performed propensity score matching analysis using various clinicopathological characteristics to reduce the impact of selection bias and potential ambiguity. Individual patient propensity scores were calculated using logistic regression analysis. Patients with mETE were matched to those with gETE at a 1:1 ratio. After propensity score matching, DFS and long-term oncological outcomes were compared between the mETE and gETE groups. DFS predictors after propensity score matching were validated using univariate and multivariate Cox regression analyses, similar to that before propensity score matching. Differences with *p*-values of less than 0.05 were considered statistically significant. All statistical analyses were performed using Statistical Package for the Social Sciences (version 24.0; IBM Corp., Armonk, NY, USA).

## 3. Results

### 3.1. Comparison of Baseline Clinicopathological Characteristics According to the ETE Status before Propensity Score Matching

The results of the comparison of the clinicopathological characteristics of each group are presented in Table 1. The extent of surgery was significantly more extensive in the gETE group than in the mETE group (*p* < 0.001). The mean tumor size of the gETE group was significantly larger than that of the mETE group (1.0 ± 0.7 cm vs. 1.8 ± 1.0 cm; *p* < 0.001). The gETE group had a significantly higher prevalence of bilaterality (29.8% vs. 39.2%; *p* = 0.003). The incidence of lymphatic, vascular, and perineural invasions was higher in the gETE group than in the mETE group (*p* < 0.001, *p* = 0.001, and *p* = 0.001, respectively). Regarding the pathological N stage, the gETE group exhibited a significantly higher grade (*p* < 0.001). RAI therapy was performed more frequently in the gETE group (73.1% vs. 91.6%; *p* < 0.001). However, no statistically significant difference in the recurrence rate was observed between the two groups (4.2% vs. 6.8%; *p* = 0.072). Moreover, BRAFV600E positivity did not significantly differ between the gETE and mETE groups (86.5% vs. 85.1%; *p* = 0.598).

### 3.2. Univariate and Multivariate Analyses of Risk Factors for Recurrence before Propensity Score Matching

Table 2 presents the results of univariate and multivariate Cox regression analyses for evaluating risk factors associated with DFS before propensity score matching. Gender, age, tumor size, mETE, gETE, multifocality, bilaterality, lymphatic invasion, vascular invasion, harvested LNs, positive LNs, T stage, N stage, and RAI therapy were identified as significant risk factors for DFS in the univariate analysis. In the multivariate analysis, the number of positive LNs (HR, 1.059; 95% CI, 1.041–1.077; *p* < 0.001), N1a stage (HR, 2.978; 95% CI, 1.785–4. 968; *p* < 0.001), N1b stage (HR, 2.341; 95% CI, 1.175–4. 662; *p* = 0.016), and RAI therapy (HR, 2.587; 95% CI, 1.498–4.468; *p* = 0.001) were significantly associated with recurrence. However, both mETE (HR, 1.362; 95% CI, 0.891–2.083; *p* = 0.154) and gETE (HR, 1.826; 95% CI, 0.984–3.520; *p* = 0.072) were not identified as risk factors for recurrence in the multivariate analysis. Kaplan–Meier survival analysis revealed a significant difference in DFS among the three groups before propensity score matching (log-rank *p* < 0.001) (Figure 1).

### 3.3. Comparison of Baseline Clinicopathological Characteristics between the mETE and gETE Groups after Propensity Score Matching

Table 3 shows the results of the comparison of the clinicopathological characteristics of the mETE and gETE groups after propensity score matching. Propensity score matching yielded 213 matched pairs of patients. After propensity score matching, there were no significant differences in clinicopathological characteristics between the matched groups. Ten (4.7%) patients in the mETE group and thirteen (6.1%) patients in the gETE group had recurrence; however, this result was not statistically significant (*p* = 0.668).

### 3.4. Univariate and Multivariate Analyses of Risk Factors for Recurrence after Propensity Score Matching

gETE was not associated with an increased risk of recurrence (HR, 1.301; 95% CI, 0.570–2.966; *p* = 0.532) compared with mETE in the univariate analysis. Only lymphatic invasion (HR, 3.694; 95% CI, 1.039–13.142; *p* = 0.044) and the number of positive LNs (HR, 1.126; 95% CI, 1.043–1.215; *p* = 0.003) were confirmed as significant predictors of recurrence (Table 4). The DFS curves of mETE and gETE after propensity score matching are illustrated using Kaplan–Meier survival analysis (Figure 2). DFS of the gETE group did not significantly differ from that of the mETE group (log-rank *p* = 0.531).

### 3.5. Sub-Analysis of Clinicopathological Characteristics According to ETE Status in PTMC

Sub-analysis of baseline clinicopathological characteristics of the patients with PTMC according to ETE status is summarized in Table 5. The mean age of the gETE group was significantly higher than that of the mETE group (51.3 ± 11.3 years vs. 47.6 ± 11.4 years; *p* = 0.018). Patients in the gETE group underwent significantly more extensive surgeries than those in the mETE group (*p* = 0.006). The mean tumor size was significantly larger in the gETE group than in the mETE group (*p* < 0.001). A significantly higher prevalence of bilaterality was observed in the gETE group than in the mETE group (37.5% vs. 24.7%; *p* = 0.046). The gETE group had significantly more advanced N stage than the mETE group (*p* = 0.017). RAI therapy was performed more frequently in the gETE group (63.0% vs. 85.7%; *p* = 0.001). However, no significant differences in the recurrence rates were found between the mETE and gETE groups (2.7% vs. 1.8%; *p* = 1.000).

Table 6 shows the risk factors for recurrence in PTMC. The number of positive LNs (HR, 1.123; 95% CI, 1.034–1.219; *p* = 0.006) and RAI therapy (HR, 3.890; 95% CI, 2.030–7.452; *p* < 0.001) were considered significant predictors of recurrence. However, both mETE (HR, 1.039; 95% CI, 0.607–1.779; *p* = 0.889) and gETE (0.522; 95% CI, 0.069–3.928; *p* = 0.527) were not identified as risk factors for recurrence in the multivariate analysis. Kaplan–Meier analysis showed that DFS did not significantly differ among the three groups (log-rank *p* = 0.065) (Figure 3).

## 4. Discussion

In this study, no significant difference in recurrence rates was observed between the mETE and gETE groups. To reduce the effects of selection bias, propensity score matching was performed to adjust for several clinicopathological characteristics between the mETE and gETE groups. Our results suggest that gETE and mETE have similar long-term oncological outcomes.

The AJCC/UICC TNM staging system is recommended for patients with DTC based on its usefulness in predicting disease prognosis. From January 2018, the eighth edition of the AJCC/UICC TNM staging system has been applied to overcome several limitations identified in its seventh edition [4]. The modification of the age cutoff from 45 to 55 years is a major change in the eighth edition. Several studies have suggested that the age of 45 years may not statistically be the cutoff value for the staging system [15,16]. Another change is a decrease in the unfavorable prognostic significance of cervical LN metastases. The definition of central neck (N1a) was expanded to include level VII in addition to level VI. Another notable change in the eighth edition is the definition of the T classification of thyroid cancer. The seventh edition of the AJCC/UICC TNM staging system classified patients with mETE as T3 [3,17]. However, in the eighth edition, mETE with tumors of ≤4 cm in size was excluded from the T3 classification. Tumors with gETE invading strap muscles only were classified as T3b [4,5]. Several patients with mETE were reclassified as T1 or T2 based on their primary tumor size in the eighth edition.

gETE was long recognized as a factor that adversely affected prognosis in PTC. Several studies have shown that gETE is closely related to risk factors for recurrence and disease-specific death [18,19]. Victoria et al. have demonstrated that ETE invading strap muscles alone (T3b) increased the risk of disease-specific death [20]. Several studies have compared mETE with gETE or no ETE in terms of whether mETE affects the prognosis of PTC. Ito et al. and Arora et al. have revealed that patients with gETE had a higher recurrence risk than those with mETE [7,9]. Subsequent studies have shown that mETE had no significant effects on local recurrence and survival [12,14,21,22]. In contrast, other studies have suggested that mETE has a prognosis similar to that in gETE [23,24]. Recently, Debora et al. have concluded that the presence of mETE should still be considered an intermediate-risk factor for recurrence, suggesting that both mETE and gETE are independent risk factors for the risk of recurrence in PTC, except for microcarcinomas without LN metastases [10]. However, the prognostic significance of mETE remains controversial.

Therefore, we compared the oncological outcomes between mETE and gETE in PTC using propensity score matching to reduce selection bias. Our data revealed a significant difference in DFS among the three groups in Kaplan–Meier survival analysis before propensity score matching (log-rank *p* < 0.001). After propensity score matching, however, the recurrence rates in the mETE and gETE groups were 4.7% and 6.1%, respectively, which were not statistically significantly different (*p* = 0.668). No significant difference in DFS (log-rank *p* = 0.531) was observed between the mETE and gETE groups. Thus, our findings suggest that patients with gETE invading strap muscles only should undergo a more conservative surgery or staging should be modified in patients in the T classification.

The BRAFV600E mutation has been identified as the most common and specific genetic mutation in PTC, with a prevalence ranging from 37% to 83% [25,26]. In this study, 79.8% of the patients had the BRAFV600E mutation. This result is consistent with those reported in previous studies. The BRAFV600E mutation is associated with more aggressive clinicopathological characteristics and a poorer prognosis of PTC [27,28]. The BRAFV600E mutation is significantly associated with ETE in patients with PTC, including PTMC [29,30]. Lee et al. have predicted ETE before surgery depending on the presence or absence of the BRAFV600E mutation [31]. Similar to the results of other studies, the BRAFV600E mutation was higher in the mETE and gETE groups than in the no ETE group in this study. However, no significant difference was observed between the mETE and gETE groups. Further studies on the BRAFV600E mutation should be conducted to clarify its correlation with ETE.

Our data suggest that LN metastasis is an independent risk factor for DFS both before and after propensity score matching. In previous studies, the presence of LN metastasis has not been regarded as a factor affecting risk stratification, which differed from other tumor factors, such as tumor size or aggressive histological features [32,33]. LN metastasis is not considered an independent factor for prognosis, although LN metastasis has prognostic importance in older patients [34,35]. In contrast, Liu et al. have reported that the recurrence and disease-specific mortality rates were higher in the LN metastasis group at the 10-year follow-up [36]. Several studies have suggested that the presence of LN metastases in PTC was an independent predictor of locoregional recurrence, distant metastases, or survival [37,38,39,40]. However, whether LN metastasis influences the recurrence and mortality rates in patients with PTC remains controversial.

In this study, the rate of receiving postoperative RAI therapy among all patients was 53.6%. The proportion of patients receiving RAI therapy was significantly higher in the gETE groups than in the mETE group (73.1% vs. 91.6%; *p* < 0.001), and a similar tendency was also observed in PTMC. According to the ATA management guidelines, RAI therapy is considered in intermediate-risk patients and is routinely recommended for high-risk patients [6]. Since RAI therapy was determined according to the risk stratification guidelines, more patients with gETE received RAI therapy. Numerous studies have reported that RAI therapy could significantly reduce recurrence and mortality in PTC [38,41]. However, it was revealed that RAI therapy was not an independent risk factor for DFS in this study. The fact that a higher proportion of patients with gETE received RAI therapy may have influenced the outcome that RAI therapy was not an independent risk factor for recurrence.

We performed a sub-analysis of ETE as a prognostic factor in patients with PTMC. The incidence of ETE in PTMC varied, ranging from 4.5% to 31.9% [42,43,44]. In this study, the incidence of mETE and gETE was 36.4% and 1.7%, respectively. The recurrence rate in patients with PTMC ranged from 3% to 16.7% [42,43,44]. The recurrence rate in a study involving patients with PTMC with mETE alone was 3.8%, which was comparable to 2.7% for the mETE group in this study [40]. Several studies have compared mETE with no ETE as a prognostic factor in patients with PTMC [13,45,46]. However, few studies have compared mETE with gETE in patients with PTMC due to the small number of ETE cases in PTMC, particularly in gETE. No significant difference in long-term oncological outcomes was observed between the mETE and gETE groups. Multicenter studies with larger samples are needed to investigate the correlation of ETE with long-term outcomes in PTMC.

This study has several limitations. First, this study adopted a retrospective single-center study design. There may be a selection bias because the data were collected at a single tertiary institution, which did not represent the entire patient population. A histological diagnosis of mETE and gETE could be variable and, to some extent, subjective, because it may vary among pathologists or surgeons. Additionally, several patients in the mETE and gETE groups, including PTMC, received RAI therapy, which may have affected recurrence or survival. Finally, the mean follow-up period was short (122.7 ± 22.5 months). Longer follow-up is necessary to determine the prognosis of patients with PTC, as it has indolent features.

The most important strength of this study is that we performed propensity score matching to adjust for differences in clinicopathological characteristics and minimize selection bias, which yielded more reliable results. Moreover, this study involved one of the largest cohorts of patients with PTC who underwent surgery (n = 4452). To the best of our knowledge, few studies have evaluated the impact of mETE and gETE on the long-term prognosis of PTMC.

## 5. Conclusions

This study demonstrated that both mETE and gETE were not independent risk factors for recurrence in PTC. This observation suggests that gETE invading strap muscles alone might not negatively affect the oncological outcomes in PTC. Our findings could affect the decision-making for patients with gETE invading strap muscles only. Further studies are required to determine the modification of gETE invading strap muscles alone in the T classification.

## Figures and Tables

**Figure 1 cancers-14-02370-f001:**
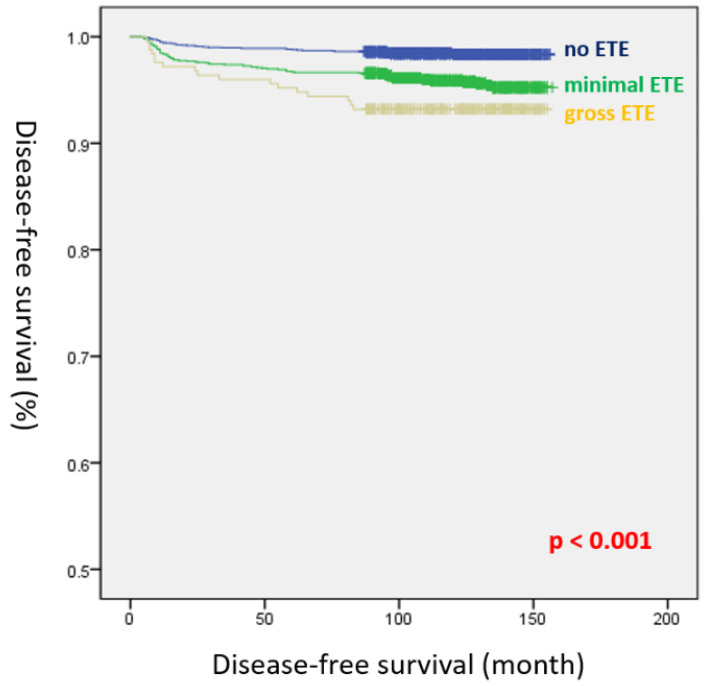
Disease-free survival curves of the three groups before propensity score matching (log-rank *p* < 0.001).

**Figure 2 cancers-14-02370-f002:**
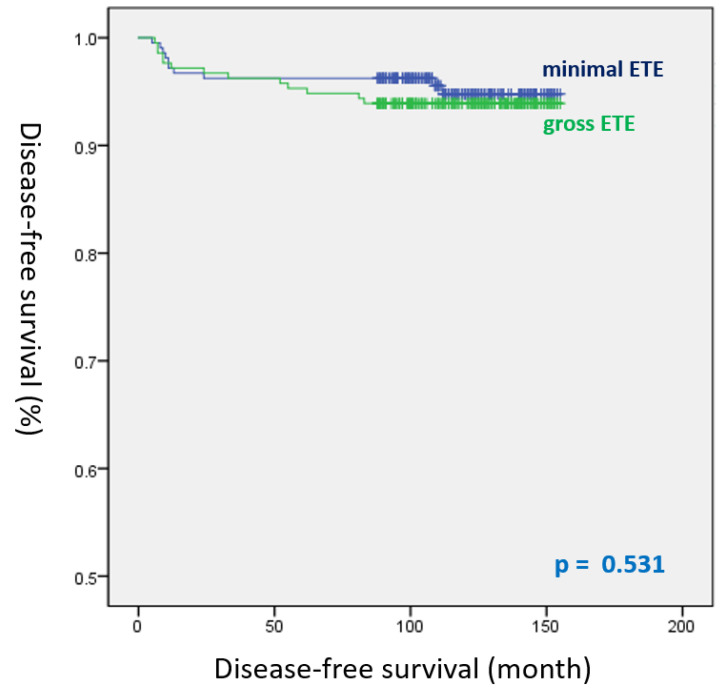
Disease-free survival curves of the mETE and gETE groups after propensity score matching (log-rank *p* = 0.531).

**Figure 3 cancers-14-02370-f003:**
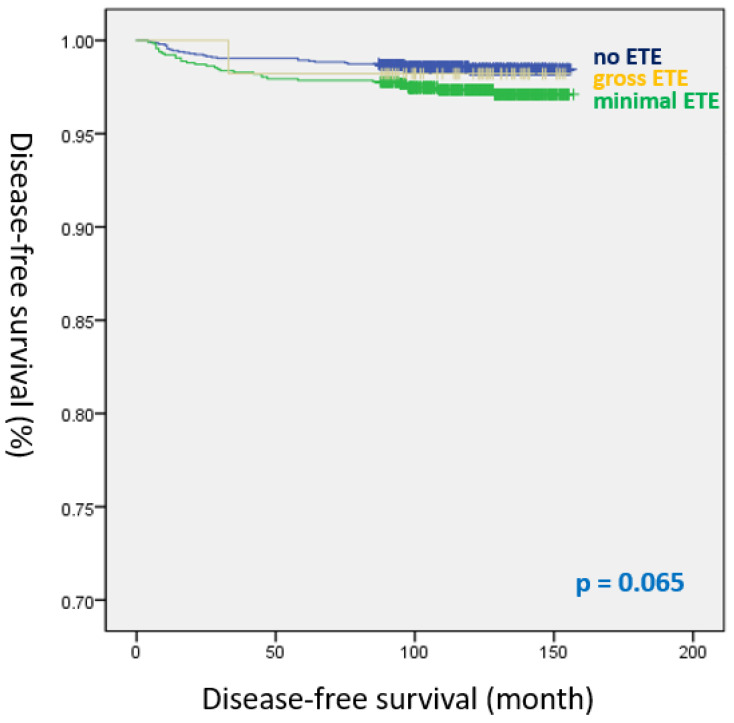
Disease-free survival curves of the three groups in papillary thyroid microcarcinoma (log-rank *p* = 0.065).

**Table 1 cancers-14-02370-t001:** Comparison of clinicopathological characteristics according to the ETE status before propensity score matching.

	No ETE (A) (n = 2411)	Minimal ETE (B) (n = 1791)	Gross ETE (C) (n = 250)	*p*-Value (A vs. B)	*p*-Value (A vs. C)	*p*-Value (B vs. C)
Age (years)	45.7 ± 11.8 (range, 13–88)	47.1 ± 12.1 (range, 12–80)	49.8 ± 13.4 (range, 11–81)	0.521	0.016	0.050
Female	1943 (80.6%)	1435 (80.1%)	208 (83.2%)	0.724	0.353	0.269
Extent of surgery				<0.001	<0.001	<0.001
Less than TT	863 (35.8%)	269 (15.0%)	5 (2.0%)			
TT and/or mRND	1548 (64.2%)	1522 (85.0%)	245 (98.0%)			
Tumor size (cm)	0.8 ± 0.6 (range, 0.2–6.0)	1.0 ± 0.7 (range, 0.2–6.0)	1.8 ± 1.0 (range, 0.2–6.0)	0.003	<0.001	<0.001
Multifocality	740 (30.7%)	822 (45.9%)	120 (48.0%)	<0.001	<0.001	0.543
Bilaterality	432 (17.9%)	534 (29.8%)	98 (39.2%)	<0.001	<0.001	0.003
Lymphatic invasion	355 (14.7%)	665 (37.1%)	146 (58.4%)	<0.001	<0.001	<0.001
Vascular invasion	16 (0.7%)	60 (3.4%)	20 (8.0%)	<0.001	<0.001	0.001
Perineural invasion	9 (0.4%)	70 (2.9%)	29 (11.6%)	<0.001	<0.001	<0.001
BRAFV600E positive	1618/2176 (74.4%)	1413/1634 (86.5%)	183/215 (85.1%)	<0.001	<0.001	0.598
Harvested LNs	8.9 ± 9.4	15.6 ± 19.5	24.7 ± 25.8	<0.001	<0.001	<0.001
Positive LNs	1.0± 2.5	3.2 ± 5.5	5.5 ± 6.8	<0.001	<0.001	<0.001
T stage				0.039	<0.001	<0.001
T1	2299 (95.4%)	1677 (93.6%)	0			
T2	99 (4.1%)	104 (5.8%)	0			
T3a	13 (0.5%)	10 (0.6%)	0			
T3b	0	0	250 (100%)			
N stage				<0.001	<0.001	<0.001
N0	1623 (67.3%)	727 (40.6%)	59 (23.6%)			
N1a	722 (29.9%)	804 (44.9%)	117 (46.8%)			
N1b	66 (2.7%)	260 (14.5%)	74 (29.6%)			
M stage				0.579	0.179	0.324
M1	1 (0.0%)	2 (0.1%)	1 (0.4%)			
TNM stage				<0.001	<0.001	<0.001
Stage I	2261 (93.8%)	1565 (87.4%)	152 (60.8%)			
Stage II	149 (6.2%)	225 (12.6%)	97 (38.8%)			
Stage IV	1 (0.0%)	1 (0.1%)	1 (0.4%)			
RAI therapy	849 (35.2%)	1309 (73.1%)	229 (91.6%)	<0.001	<0.001	<0.001
Recurrence	38 (1.6%)	75 (4.2%)	17 (6.8%)	<0.001	<0.001	0.072

Data are expressed as number of patients (%), or mean ± standard deviation. A statistically significant difference was defined as *p* < 0.05. Abbreviation: ETE, extrathyroidal extension; TT, total thyroidectomy; mRND, modified radical neck dissection; LN, lymph node; T, tumor; N, node; M, metastasis; RAI, radioactive iodine.

**Table 2 cancers-14-02370-t002:** Univariate and multivariate analyses of risk factors for recurrence before propensity score matching.

	Univariate		Multivariate	
	HR (95% CI)	*p*-Value	HR (95% CI)	*p*-Value
Gender				
Female	ref.			
Male	1.610 (1.097–2.365)	0.015		
Age (years)				
>45	ref.			
≤45	1.676 (1.182–2.378)	0.004		
Tumor size				
≤1cm	ref.			
>1cm	2.944 (2.086–4.154)	<0.001		
ETE	3.409 (2.140–5.432)	<0.001		
No ETE	ref.		ref.	
Minimal ETE	2.662 (1.802–3.934)	<0.001	1.362 (0.891–2.083)	0.154
Gross ETE	4.350 (2.455–7.708)	<0.001	1.826 (0.984–3.520)	0.072
Multifocality	1.689 (1.197–2.382)	0.003		
Bilaterality	1.717 (1.196–2.464)	0.003		
Lymphatic invasion	3.832 (2.708–5.423)	<0.001		
Vascular invasion	3.063 (1.498–6.263)	0.002		
Harvested LNs	1.021 (1.015–1.026)	<0.001		
Positive LNs	1.074 (1.062–1.086)	<0.001	1.059 (1.041–1.077)	<0.001
T stage				
T1	ref.			
T2	3.191 (1.851–5.501)	<0.001		
T3a	5.624 (1.782–17.749)	0.003		
T3b	2.870 (1.713–4.809)	<0.001		
N stage				
N0	ref.		ref.	
N1a	5.216 (3.246–8.384)	<0.001	2.978 (1.785–4.968)	<0.001
N1b	9.010 (5.236–15.506)	<0.001	2.341 (1.175–4.662)	0.016
RAI therapy	5.332 (3.241–8.774)	<0.001	2.587 (1.498–4.468)	0.001

Data are expressed as hazard ratio (HR) and 95% confidence interval (CI). A statistically significant difference was defined as *p* < 0.05. Abbreviations: ETE, extrathyroidal extension; LN, lymph node; T, tumor; N, node; RAI, radioactive iodine.

**Table 3 cancers-14-02370-t003:** Comparison of clinicopathological characteristics between the minimal ETE and gross ETE groups after propensity score matching.

	Minimal ETE (n = 213)	Gross ETE (n = 213)	*p*-Value
Age (years)	49.5 ± 12.3 (range, 19–74)	48.6 ± 12.4 (range, 19–74)	0.484
Female	162 (76.1%)	177 (83.1%)	0.092
Extent of surgery			1.000
Less than TT	4 (1.9%)	4 (1.9%)	
TT and/or mRND	209 (98.1%)	209 (98.1%)	
Tumor size (cm)	1.6 ± 0.9 (range, 0.2–5.0)	1.6 ± 0.8 (range, 0.2–5.0)	0.390
Multifocality	119 (55.9%)	103 (48.4%)	0.146
Bilaterality	85 (39.9%)	84 (39.4%)	1.000
Lymphatic invasion	122 (57.3%)	117 (54.9%)	0.696
Vascular invasion	11 (5.2%)	11 (5.2%)	1.000
Perineural invasion	16 (7.5%)	20 (9.4%)	0.601
BRAFV600E positive			
Harvested LNs	23.0 ± 25.2	22.4 ± 22.3	0.801
Positive LNs	5.0 ± 7.2	5.2 ± 6.5	0.810
T stage			<0.001
T1	163 (76.5%)	0 (0.0%)	
T2	44 (20.7%)	0 (0.0%)	
T3a	6 (2.8%)	0 (0.0%)	
T3b	0 (0.0%)	213 (100.0%)	
N stage			0.990
N0	48 (22.5%)	47 (22.1%)	
N1a	110 (51.6%)	110 (51.6%)	
N1b	55 (25.8%)	56 (26.3%)	
M stage			1.000
M1	1 (0.5%)	0 (0.0%)	
TNM stage			0.589
Stage I	137 (64.3%)	140 (65.7%)	
Stage II	75 (35.2%)	73 (34.3%)	
Stage IV	1 (0.5%)	0 (0.0%)	
RAI therapy	195 (91.5%)	196 (92.0%)	1.000
Recurrence	10 (4.7%)	13 (6.1%)	0.668

Data are expressed as number of patients (%), or mean ± standard deviation. A statistically significant difference was defined as *p* < 0.05. Abbreviation: ETE, extrathyroidal extension; TT, total thyroidectomy; mRND, modified radical neck dissection; LN, lymph node; T, tumor; N, node; M, metastasis; RAI, radioactive iodine.

**Table 4 cancers-14-02370-t004:** Univariate and multivariate analyses of risk factors for recurrence after propensity score matching.

	Univariate		Multivariate	
	HR (95% CI)	*p*-Value	HR (95% CI)	*p*-Value
Gender				
Female	ref.			
Male	3.139 (1.376–7.162)	0.007		
Age (years)				
>45	ref.			
≤45	1.123 (0.486–2.593)	0.787		
Tumor size				
≤1cm	ref.			
>1cm	3.702 (0.868–15.788)	0.077		
ETE				
Minimal ETE	ref.			
Gross ETE	1.301 (0.570–2.966)	0.532		
Multifocality	2.151 (0.885–5.228)	0.091		
Bilaterality	2.015 (0.884–4.596)	0.096		
Lymphatic invasion	5.511 (1.637–18.550)	0.006	3.694 (1.039–13.142)	0.044
Vascular invasion	0.814 (0.110–6.039)	0.841		
Harvested LNs	1.013 (1.001–1.025)	0.037		
Positive LNs	1.070 (1.037–1.105)	<0.001	1.126 (1.043–1.215)	0.003
T stage				
T1	ref.			
T2	6.516 (1.557–27.268)	0.010		
T3a	19.596 (3.273–117.313)	0.001		
T3b	3.361 (0.958–11.795)	0.058		
N stage				
N0	ref.			
N1a	1.809 (0.510–6.412)	0.358		
N1b	2.323 (0.616–8.757)	0.213		
RAI therapy	1.947 (0.262–14.444)	0.515		

Data are expressed as hazard ratio (HR) and 95% confidence interval (CI). A statistically significant difference was defined as *p* < 0.05. Abbreviations: ETE, extrathyroidal extension; LN, lymph node; T, tumor; N, node; RAI, radioactive iodine.

**Table 5 cancers-14-02370-t005:** Sub-analysis of clinicopathological characteristics according to ETE status in PTMC.

	No ETE (A) (n = 1981)	Minimal ETE (B) (n = 1167)	Gross ETE (C) (n = 56)	*p*-Value (A vs. B)	*p*-Value (A vs. C)	*p*-Value (B vs. C)
Age (years)	45.9 ± 11.4 (range, 16–88)	47.6 ± 11.4 (range, 20–80)	51.3 ± 11.3 (range, 27–74)	<0.001	<0.001	0.018
Female	1616 (81.6%)	967 (82.9%)	45 (80.4%)	0.389	0.955	0.761
Extent of surgery				<0.001	<0.001	0.006
Less than TT	782 (39.5)	253 (21.7%)	3 (5.4%)			
TT and/or mRND	1199 (60.5%)	914 (78.3%)	53 (94.6%)			
Tumor size (cm)	0.6 ± 0.2	0.7 ± 0.2	0.8 ± 0.2	<0.001	<0.001	<0.001
Multifocality	578 (29.2%)	492 (42.2%)	30 (53.6%)	<0.001	<0.001	0.122
Bilaterality	320 (16.2%)	288 (24.7%)	21 (37.5%)	<0.001	<0.001	0.046
Lymphatic invasion	254 (12.8%)	333 (28.5%)	22 (39.3%)	<0.001	<0.001	0.114
Vascular invasion	4 (0.2%)	23 (2.0%)	1 (1.8%)	<0.001	0.321	1.000
Perineural invasion	8 (0.4%)	31 (2.7%)	4 (7.1%)	<0.001	<0.001	0.120
BRAFV600E positive	1375/1783 (77.1%)	910/1060 (85.8%)	38/50 (76.0%)	<0.001	0.988	0.085
Harvested LNs	8.2 ± 8.2	11.1 ± 13.1	18.9 ± 20.6	<0.001	<0.001	0.007
Positive LNs	0.8 ± 2.0	1.8 ± 3.2	3.2 ± 4.9	<0.001	<0.001	0.035
T stage				1.000	<0.001	<0.001
T1	1981 (100.0%)	1167 (100.0%)	0			
T2	0	0	0			
T3a	0	0	0			
T3b	0	0	56 (100%)			
N stage				<0.001	<0.001	0.017
N0	1382 (69.8%)	586 (50.2%)	23 (41.1%)			
N1a	563 (28.4%)	494 (42.3%)	23 (41.1%)			
N1b	36 (1.8%)	87 (7.5%)	10 (17.9%)			
M stage				0.789	NA	1.000
M1	0 (0.0%)	1 (0.1%)	0 (0.0%)			
TNM stage				<0.001	<0.001	<0.001
Stage I	1877 (94.8%)	1056 (90.5%)	31 (55.4%)			
Stage II	104 (5.2%)	110 (9.4%)	25 (44.6%)			
Stage IV	0 (0.0%)	1 (0.1%)	0 (0.0%)			
RAI therapy	566 (28.6%)	735 (63.0%)	48 (85.7%)	<0.001	<0.001	0.001
Recurrence	29 (1.5%)	31 (2.7%)	1 (1.8%)	0.026	1.000	1.000

Data are expressed as number of patients (%), or mean ± standard deviation. A statistically significant difference was defined as *p* < 0.05. Abbreviation: ETE, extrathyroidal extension; PTMC, papillary thyroid microcarcinoma; TT, total thyroidectomy; mRND, modified radical neck dissection; LN, lymph node; T, tumor; N, node; M, metastasis; RAI, radioactive iodine; NA, not applicable.

**Table 6 cancers-14-02370-t006:** Univariate and multivariate analyses of risk factors for recurrence in PTMC.

	Univariate		Multivariate	
	HR (95% CI)	*p*-Value	HR (95% CI)	*p*-Value
Gender				
Female	ref.			
Male	1.377 (0.758–2.501)	0.293		
Age (years)				
>45	ref.			
≤45	1.684 (1.011–2.804)	0.045		
Tumor size				
≤1cm				
>1cm				
ETE				
No ETE	ref.		ref.	
Minimal ETE	1.813 (1.092–3.008)	0.021	1.039 (0.607–1.779)	0.889
Gross ETE	1.208 (0.165–8.866)	0.853	0.522 (0.069–3.928)	0.527
Multifocality	1.663 (1.005–2.753)	0.048		
Bilaterality	1.124 (0.609–2.075)	0.708		
Lymphatic invasion	3.238 (1.949–5.379)	<0.001		
Vascular invasion	3.892 (0.951–15.929)	0.059		
Harvested LNs	1.017 (1.001–1.034)	0.035		
Positive LNs	1.133 (1.088–1.181)	<0.001	1.123 (1.034–1.219)	0.006
T stage				
T1	ref.			
T2				
T3a				
T3b	0.975 (0.505–1.885)	0.941		
N stage				
N0	ref.			
N1a	4.340 (2.457–7.678)	<0.001		
N1b	4.476 (1.290–3.497)	0.003		
RAI therapy	5.020 (2.719–9.269)	<0.001	3.890 (2.030–7.452)	<0.001

Data are expressed as hazard ratio (HR) and 95% confidence interval (CI). A statistically signifi-cant difference was defined as *p* < 0.05. Abbreviations: PTMC, papillary thyroid microcarcinoma; ETE, extrathyroidal extension; LN, lymph node; T, tumor; N, node; RAI, radioactive iodine.

## Data Availability

The original contributions presented in the study are included in the article. Further inquiries can be directed to the corresponding author.

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
