# Peer review of "Is Gross Extrathyroidal Extension to Strap Muscles (T3b) Only a Risk Factor for Recurrence in Papillary Thyroid Carcinoma? A Propensity Score Matching Study"

_cancers, 2022, doi:10.3390/cancers14102370_

Round 1

Reviewer 1 Report

This is a retrospective comparatory study that aimed to assess the impact of strap musculature involvement in papillary thyroid carcinoma on survival. 

The objectives of the study are presented clearly and the introduction section communicates the need for investigating the impact of extra-thyroid tumor spread on survival.

The Materials and Methods section has clearly defined inclusion and exclusion criteria, and has listed adequate statistical tests.

The Results section reports an important clinical fact, that ENE has little impact on disease outcome,  but an already known fact, not adequately addressed in literature, and corrected with this study.

The authors have addressed all major issues in the discussion section. 

I would congratulate the authors on a well-crafted study and would endorse further consideration.

Author Response

Response; I sincerely appreciate your review and comments on our study.

Thanks to your in-depth review, if our research is published in the journal Cancers, it is hoped that it will contribute to a better understanding of papillary thyroid cancer.

Reviewer 2 Report

This study involved a large cohort of PTC patients in an attempt to clarify the prognostic significance of gross extra-thyroidal extension. Only minor issues are raised:
A relatively high percentage of patients has been submitted to less than total thyroidectomy. I think that a subgroup analysis of patients submitted to total and less that total thyroidectomy would add to the significance of the study.

Author Response

Response; I sincerely appreciate for your detailed review.

Since mETE is confirmed by pathologic findings after surgery, while gETE is grossly confirmed intraoperatively, the proportion of patients with less than total thyroidectomy may be relatively higher in mETE group than gETE group (85 vs. 98%).

I think you suggested that it would be better to do a subgroup analysis because the ratio of the extent of surgery between the two groups is different. As shown in Table 3, we adjusted for a difference in extent for surgery through propensity score matching rather than subgroup analysis for the correlation between the extent of surgery and oncologic outcomes in the two groups (98.1 vs. 98.1%). As a results, there was no statistically significant differences with respect to the recurrence rate between two groups.

Reviewer 3 Report

This is a retrospective single center study evaluating the clinicopathological characteristics and recurrence rates between minimal and gross extrathyroidal extension in case of papillary thyroid carcinoma.

It was a pleasure to read this article that appears very well made and written.

The conclusions of the authors are convincing demonstrating that both mETE and gETE were not independent risk factors for recurrence in PTC.

I find this article aqppropriate for publication without need of major revision

Author Response

Response; I sincerely appreciate your detailed review and comments on our study.

Thanks to your in-depth review, if our research is published in the journal Cancers, it is hoped that it will contribute to a better understanding of papillary thyroid cancer.
